# Relationship between Apical Periodontitis and Metabolic Syndrome and Cardiovascular Events: A Cross-Sectional Study

**DOI:** 10.3390/jcm9103205

**Published:** 2020-10-04

**Authors:** Beatriz González-Navarro, Juan José Segura-Egea, Albert Estrugo-Devesa, Xavier Pintó-Sala, Enric Jane-Salas, Mari Carmen Jiménez-Sánchez, Daniel Cabanillas-Balsera, José López-López

**Affiliations:** 1Department of Odontostomatolgy, School of Medicine and Health Sciences (Dentistry)—Dental Hospital, University of Barcelona, 08970 L’Hospitalet de Llobregat, Barcelona, Spain; beatrizgonzaleznavarro@gmail.com (B.G.-N.); albertestrugo@ub.edu (A.E.-D.); enjasa19734@gmail.com (E.J.-S.); 2Oral Health and Masticatory System Group, Bellvitge Biomedical Research Institute, (IDIBELL), 08970 L’Hospitalet de Llobregat, Barcelona, Spain; 3Department of Stomatology, School of Dentistry, University of Sevilla, Calle Avicena s/n, 41009 Sevilla, Spain; jimenezsanchez6@gmail.com (M.C.J.-S.); danielcaba@gmail.com (D.C.-B.); 4Vascular Risk Unit, Department of Internal Medicine, Bellvitge University Hospital, 08970 L’Hospitalet de Llobregat, Barcelona, Spain; xpinto@bellvitgehospital.cat; 5Clinical Head of the Odontological Hospital University of Barcelona, University of Barcelona, 08970 L’Hospitalet de LLobregat, Barcelona, Spain

**Keywords:** apical periodontitis, cardiovascular disease, cardiovascular events, endodontic infection, metabolic syndrome, periapical index, total dental index

## Abstract

Aim: Both apical periodontitis (AP) and metabolic syndrome (MetS) are associated with atherosclerotic cardiovascular disease (ACVD), the main cause of cardiovascular events. The aim of this study was to investigate the prevalence of AP and the oral inflammatory burden in control subjects and patients suffering cardiovascular events, analyzing the possible association between AP and the oral inflammatory burden with MetS. Materials and Methods: Using a cross-sectional design, 83 patients suffering a cardiovascular event were recruited in the study group (SG), and 48 patients without cardiovascular events were included in the control group (CG). Periapical index (PAI) was used to diagnose AP, and total dental index (TDI) was used to assess the total oral inflammatory burden. Diagnosis of MetS was made by meeting three or more American Heart Association Scientific Statement components. Results: In the multivariate logistic regression analysis, the number of teeth with AP (OR = 2.3; 95% C.I. = 1.3–4.3; *p* = 0.006) and TDI scores (OR = 1.5; 95% C.I. = 1.2–1.9; *p* = 0.001), significantly correlated with cardiovascular events. MetS was strongly associated (OR = 18.0; 95% C.I. = 6.5–49.7; *p* = 0000) with cardiovascular events. Higher TDI scores were significantly associated with MetS (OR = 1.3; 95% C.I. = 1.1–1.6; *p* = 0.003. Neither the number of root-filled teeth (RFT) (OR = 0.9; 95% C.I. = 0.6–1.3; *p* = 0.61) nor the number of teeth with apical periodontitis (OR = 1.1; 95% C.I. = 0.8–1.7; *p* = 0.49) were associated with MetS. Conclusions: Apical periodontitis is significantly associated with cardiovascular events. Total oral inflammatory burden assessed by TDI, but not AP alone, is associated with MetS.

## 1. Introduction

Apical periodontitis (AP) is the inflammatory response of periapical tissues caused by the leakage of polymicrobial and antigenic content of the root canal system through the apical foramen [1]. AP affects 34% to 61% of people [2,3], and is especially associated with root-filled teeth (RFT) [4,5]. The weighted average of teeth with AP in adult patients is 5.4%, with 36% of RFT associated with radiolucent periapical lesions [6], the radiographic sign of AP [7]. The possible relationship of AP with systemic pathologies has been raised [8], having found an association with diabetes [9] and cardiovascular disease [10].

On the other hand, metabolic syndrome (MetS) is a group of metabolic disorders characterized by abdominal obesity, hyperglycemia, hypertriglyceridemia, hypertension, and low levels of high-density lipoprotein cholesterol (HDL-c) [11]. Approximately 20–30% of the world population presents MetS [12], but the percentages may vary according to the classification used for diagnosis, and according to the study population [11,13]. MetS increases the risk of atherosclerotic cardiovascular disease (ACVD), a pathology of multifactorial origin caused by the accumulation of lipids in arterial walls [14]. ACVD is associated with a high incidence of coronary artery disease, atherothrombotic accidents, and cardiovascular events such as myocardial infarction, cardiomyopathy, arrhythmias, stroke, and cardiac arrest [15]. These cardiovascular events make ACVD the leading cause of death in Europe, the United States, and most of Asia [16].

Recently, an experimental study conducted in rats has found an association between AP and atherosclerosis, the major contributor to ACVD [17]. Moreover, several observational studies have reported statistically significant excess risk for ACVD and cardiovascular events in patients with oral inflammation [18,19] including periodontal disease [15] and AP [20,21,22,23,24,25,26,27]. AP has also been associated with poor endothelial flow reserve and early endothelial dysfunction in young adults [28]. Conversely, treatment of AP through root canal treatment (RCT) has been shown to improve systemic inflammation, ameliorating early endothelial dysfunction [29]. In addition, it has been reported that having RFT is associated with a lower probability of suffering coronary heart disease and a lower risk of mortality from cardiovascular events [30]. Despite all these findings, the causal association between oral infections and ACVD cannot be considered proven [10,31].

Both AP and MetS are associated with cardiovascular events. However, no studies have analyzed the relationship between MetS and AP, which could at least partly explain the association of AP with ACVD. This is why the two aims of this cross-sectional study are: (1) to investigate the prevalence of AP and the oral inflammatory burden in control patients and patients suffering cardiovascular events; and (2) to investigate the prevalence of AP and TDI scores in patients with and without metabolic syndrome. The null hypotheses were that the prevalence of AP and the TDI score were the same in patients with cardiovascular events and in the control subjects as well as in patients with or without metabolic syndrome.

## 2. Methodology

All participants provided written informed consent upon entering the study, and the study was approved by the Ethics Committee of the Bellvitge University Hospital Research Institute (IDIBELL—Reference: PR187/15).

### 2.1. Patients and Study Design

Patients (*n* = 131) were recruited from those who were treated at Bellvitge University Hospital between 2016–2018, with a maximum duration of three months after having suffered an atherothrombotic accident/cardiovascular event (acute myocardial infarction, unstable angina, ischemic stroke, or lower limb arterial disease) (Study Group, SG, *n* = 83). The inclusion criteria were: ages between 35 and 60 years old, having eight or more teeth in the mouth, and agreeing to undergo an intraoral examination and an orthopantomography (OPG). Exclusion criteria were having less than eight teeth in the mouth, and not agreeing to undergo the oral or radiographic examinations.

The control group (CG, *n* = 48) was made up of patients from 35 to 60 years old, who had not suffered any cardiovascular event and had attended their first visit at the Odontology Hospital University of Barcelona [HOUB]. The inclusion and exclusion criteria were the same.

The patients included in both groups underwent a detailed medical history, an oral examination, a blood test with metabolic syndrome factors, and an orthopantomography (OPG). The blood tests were performed at the University Hospital of Bellvitge and the OPG was carried out in the Odontology Hospital of the University of Barcelona.

All of the patients in the study and the control group agreed to do blood analyses, and signed an informed consent.

### 2.2. Metabolic Syndrome Diagnosis

Diagnosis of MetS was based on the following five components: central obesity (waist circumference ≥102 cm for men and ≥88 cm for women); hypertriglyceridemia (serum triglyceride ≥150 mg/dL or on medication); low HDL cholesterol (<40 mg/dL for men and <50 mg/dL for women or on medication); high blood pressure (systolic: ≥130 mmHg or diastolic: ≥85 mmHg or on medication); and high fasting plasma glucose (≥100 mg/dL or on medication) [11].

Waist circumference was measured midway between the inferior margin of the ribcage and the iliac crest horizontally using a measuring tape. Blood pressure was measured at three consecutive times and the average value was used for the analysis. Twelve hour fasting blood sample was collected from the antecubital vein and was analyzed for serum cholesterol, triglycerides, and fasting plasma glucose.

### 2.3. Radiographic Examination and Periapical Status Assessment

Radiographic periapical status was diagnosed using digital orthopantomographies taken by two proficient technicians with more than a decade of experience (Promax, Planmeca, class 1, type B, 80 KHz, Planmeca, Helsinki, Finland). All teeth, excluding third molars, were evaluated.

The periapical index (PAI) score [7] was used to evaluate the status of the periapical tissues. AP was diagnosed when a score higher than two was obtained (PAI ≥ 3). In the case of multi-rooted teeth, the highest score obtained in the analyzed roots was taken as the PAI score of that tooth. The teeth were classified as RFT if a radiopaque material was evident in the root canal(s). For each subject, a structured form was fulfilled containing the following information: the total number of teeth, the number and location of all teeth associated with PAI score ≥ 3, the number and location of RFT, and the number and location of RFT with PAI score ≥ 3.

To assess the total oral inflammatory status, the modified total dental index (TDI) was calculated recording all signs of infections from the intraoral examination and orthopantomography [32]. The modified TDI scale is from 0 to 10 in recording decays, periodontal disease valued with probing depth, apical periodontitis, and furcation lesions; a higher score reflects a greater number of infectious dental diseases, and consequently, a higher total infection burden of the mouth (Table 1).

### 2.4. Observers’ Calibration

Two blinded observers, expert in the evaluation of radiographs for endodontic diagnosis, examined the images and established the periapical diagnosis. Previously, these observers were calibrated in the PAI system, performing a course consisting of the analysis of 100 dental radiographic images. The assessed teeth were classified as one of the five categories within the PAI scale [7], and compared with the “gold standard atlas” and Cohen’s Kappa was calculated (0.72 to 0.79).

To determine the intra-observer reproducibility, both examiners scored the radiographs of 20 patients. After four weeks, the examiner was recalibrated in the PAI system and scored the same 20 panoramic radiographs again. The Cohen’s Kappa for the intra-observer agreement ranged from 0.84 to 0.89.

Inter-observer reproducibility was likewise determined matching the PAI scores on the 20 radiographs delivered by every observer. The agreement test gave a Cohen’s Kappa ranging from 0.78–0.86. The consensus radiographic standard was the simultaneous interpretation by the two examiners.

### 2.5. Statistical Analysis

The minimal sample size (*n* = 103) was determined using the sample size calculator software of the National Center for Advancing Translational Sciences (NIH, UK) (www.sample-size.net/sample-size-proportions) for the comparison of proportions in two independent samples, with continuity correction. A 2-sided significance level of 5% (α = 0.05, Zα = 1.960) and 80% power (β = 0.20, Zβ = 0.842) was taken into account to detect an assumed disparity among the proportion of two groups of 30 points (prevalence of AP described previously in Spanish population ∼30%; [3] hypothesized prevalence of AP in SG was 60%). Raw data were entered into Excel (Microsoft Corporation, Redmond, WA, USA). All analyses were performed in an SPSS environment (Version 11; SPSS, Inc, Chicago, IL, USA). The significance of differences among groups were determined using the χ^2^ test, the Student´s *t*-test (Welch’s *t*-test, or unequal variances t-test), the U–Mann–Whitney test, and both univariate and multivariate logistic regression analyses. Data were reported as mean ± standard deviation. According to the established significance level, a *p* value ≤ 0.05 was considered statistically significant.

## 3. Results

Table 2 shows the characteristics and dental status of the patients (*n* = 131) of both SG and CG. The mean age of the total sample was 50.0 ± 6.1 years, involving 93 men (71%) and 38 women (29%). No significant differences in age or gender were found between the two groups (*p* > 0.05). The value obtained was OR = 15.6 (95% CI = 6.3–39.2; *p* = 0.0000). There were also no significant differences between SG and CG in relation to diabetes and smoking habits (*p* > 0.05). In contrast, hypertension, body mass index, waist circumference, hypertriglyceridemia, low c-HDL, and fasting plasma glucose levels were significantly higher in the SG (*p* < 0.05). In the SG, 63 patients (76%) were diagnosed with MetS, while in the CG only eight (17%) were diagnosed (*p* < 0.0001).

No significant differences were found between SG and CG in the number of RFT (<0.05). However, the number of teeth with apical periodontitis was significantly greater in the SG (1.0 ± 1.2) than in the CG (0.4 ± 0.7) (*p* = 0.02). In the SG, 39 patients (47%) had at least one tooth with AP, whereas this percentage was lower (25%) in the CG (*p* = 0.013). Univariate logistic regression analysis, run with the number of teeth with AP as the independent variable and cardiovascular events as the dependent variable (0 = absent, 1 = present), provided an OR = 2.1 (95% C.I. = 1.3–3.3; *p* = 0.0027).

The median value of the TDI, which assessed the total oral inflammatory status, was 3 in the total sample, with an interquartile range of 1 to 5. In the SG, the median value of the TDI was 4, four times greater than that found in the CG (1) (*p* < 0.0000). Univariate logistic regression analysis, run with a TDI score as an independent variable and cardiovascular events as the dependent variable (0 = absent, 1 = present), provided an OR = 1.7 (95% C.I. = 1.4–2.1; *p* = 0.0000).

To further investigate whether cardiovascular events were related to AP, multivariate logistic regression was run with age, gender, number of teeth, smoking habits, diabetic status, metabolic syndrome, number of RFT, and number of teeth with AP as independent explanatory variables, and cardiovascular events (0 = absent, 1 = present) as the dependent variable (Table 3). The number of teeth with AP (OR = 2.3; 95% C.I. = 1.3–4.3; *p* = 0.006) significantly correlated with the prevalence of cardiovascular events (Table 3). MetS was, as expected, strongly associated (OR = 18.0; 95% C.I. = 6.5–49.7; *p* = 0000) with cardiovascular events.

Similarly, to investigate whether cardiovascular events were related to TDI, multivariate logistic regression were run with age, gender, number of teeth, smoking habits, diabetic status, metabolic syndrome, number of RFT, and TDI score as independent explanatory variables, and cardiovascular events (0 = absent, 1 = present) as the dependent variable (Table 4). TDI significantly correlated with the prevalence of cardiovascular events (OR = 1.5; 95% C.I. = 1.2−1.9; *p* = 0.001).

Multivariate logistic regression analysis was also used to assess the relationship of AP with MetS (Table 5). The analysis included as independent variables age, gender, number of teeth, diabetes, smoking habits, number of RFT, and number of teeth with AP, with MetS (0 = absent, 1 = present) the dependent variable. Neither the number of RFT (OR = 0.9; 95% C.I. = 0.6–1.3; *p* = 0.61) nor the number of teeth with apical periodontitis (OR = 1.1; 95% C.I. = 0.8–1.7; *p* = 0.49), were associated with MetS.

Finally, to investigate whether TDI scores were associated with metabolic syndrome, multivariate regression analysis was run including age, gender, number of teeth, diabetes, smoking habits, number of RFT, and TDI score as independent co-variates, and metabolic syndrome (0 = absent, 1 = present) as the dependent variable (Table 6). Higher TDI score was significantly associated with MetS (OR = 1.3; 95% C.I. = 1.1–1.6; *p* = 0.003).

## 4. Discussion

This cross-sectional study investigated the relationship of AP and total oral inflammatory burden with cardiovascular events as well as the association between these two variables and MetS. The results showed a significant association between oral infection, specifically AP, and cardiovascular events. Regarding the second objective, the total oral inflammatory burden was associated with MetS, but no significant association was found between AP and MetS.

The recruitment method of the patients was similar to that used in previous studies [33,34,35,36]. Apical periodontitis was diagnosed using the ‘periapical index’ (PAI) scoring system [7], widely used in epidemiological and clinical studies in which the presence of radiolucent periapical lesions is assessed to determine the prevalence of AP [3,37,38]. Although there may be an underestimation of lesions when panoramic radiography was used [39], the difference with periapical radiography is not statistically significant [40]. Both periapical and panoramic radiography can be used to assess periapical status, but the fact that panoramic radiography shows all teeth, together with low exposure to ionizing radiation [40], the convenience of panoramic radiographs, and the speed with which they can be obtained are advantageous compared to periapical full-mouth radiographs [41]. On the other hand, although PAI has been described for periapical radiographs [7], it has been used by numerous epidemiological studies in which the periapical status is assessed with panoramic radiographs [42,43]. The possibility of comparisons between studies carried out with calibrated observers makes this system attractive [44] to better standardize the evaluations and allow comparison with the results of other researchers. Additionally, although the PAI score system was first described for periapical radiographs [26], it has been used in many epidemiological studies for panoramic radiographs [22,24,27,37,38,45,46]. The possibility of comparisons among the studies carried out with calibrated observers makes this system attractive [41], in order to better standardize the evaluations and allow the comparison with the result of other investigators.

The total oral inflammatory burden was determined using the modified TDI [32], widely used in studies investigating the association between oral infections and cardiovascular diseases [19,24,25,45].

The results showed a significant association between the total oral inflammatory burden and cardiovascular events. A high score on the TDI is associated with a 1.5-fold (OR = 1.5; *p* = 0.001) increase in the probability of suffering a cardiovascular event. This result is consistent with those obtained in previous studies [18,32,46,47,48].

The results of the present study also found a significant association between the number of teeth with AP and cardiovascular events. A high number of teeth with AP increases the probability of suffering a cardiovascular event more than twice (OR = 2.3; *p* = 0.006). This result confirms and is consistent with the findings of previous cross-sectional studies, which found a significant association between the prevalence of AP and acute myocardial infarction (AMI) [49], or first attack of angina pectoris or AMI [2]. In addition, they agree with the studies that have found a higher prevalence of AP in patients with ACVD [25,50,51,52,53,54]. Although few longitudinal studies have been conducted investigating the relationship between AP and cardiovascular events [10], it has been found that in patients <40 years old, incident of radiolucent periapical lesions was significantly related to the development of coronary heart disease [20]. Another study showed that acute coronary syndrome is more frequent in subjects with untreated teeth with AP (OR = 2.72; *p* = 0.022) [24].

A recent study using epidemiological and genetic approaches including more than two million patients, found significant association between the presence of endodontic pathology and a history of cardiovascular events [55]. However, it should also be mentioned that several epidemiological studies did not find an association between AP and ACVD [56,57,58,59].

However, the results of the present study regarding the association between AP and cardiovascular events could be biased by the presence of periodontal disease. Taking into account that the aim of the study was not only to investigate the prevalence of AP, but also the oral inflammatory burden, assessed with TDI, periodontal patients could not be excluded from the study. Periodontal status is one of the components of total dental index (TDI) and contribute to the oral inflammatory burden. Even so, in multivariate logistic regression analysis, AP remained significantly associated with cardiovascular events.

The prevalence of RFT was not associated with cardiovascular events. This result is in accordance with previous findings [58], although other studies have found an association between endodontic status and the risk of cardiovascular event, both in the sense of increasing it [21,60] or decreasing it [30].

Unsurprisingly, MetS patients have a higher frequency of cardiovascular events, MetS being strongly associated (OR = 18.0; *p* = 0000) with ACVD. Likewise, all metabolic alterations characteristic of MetS such as abdominal obesity, hyperglycemia, hypertriglyceridemia, hypertension, and low levels of high-density lipoprotein cholesterol (HDL-c) have been associated with cardiovascular events. It is well known that MetS increases the risk of ACVD and cardiovascular events [11,14].

Regarding the possible association between oral infections and MetS, until now, no studies have addressed the relationship between total oral inflammatory burden or endodontic infection and MetS. The results of the present study show higher TDI scores significantly associated with MetS (OR = 1.3; *p* = 0.003), but neither periapical status (*p* = 0.49) nor endodontic status (*p* = 0.61) were associated with MetS. Given that endodontic infection is included in the TDI score, these results are not conclusive regarding the association or not of AP with MetS, but suggest that both AP and MetS are independently associated with cardiovascular events.

Several epidemiological studies, the majority cross-sectional, have investigated the association between periodontal disease and MetS [53], finding that the odds of periodontal disease increase with the number of MetS components present in an individual [61,62]. A study including 8314 participants found an OR = 1.4 (*p* < 0.05) for the association between periodontal disease and MetS [63]. A case-control study recently published found that patients with MetS presented worse periodontal status and higher prevalence, severity, and extension of periodontitis [64]. Dental caries, a component of TDI, has also been associated with MetS among the middle-aged urban Chinese population [65], and a recent study found an association between decayed teeth and MetS [66]. On the other hand, an association has been found between AP and lipid profile [17] as well as with MetS, as indicated by the induction of overweight and hyperglycemia in rats [67]. Moreover, association of some of the MetS components with AP such as hyperglycemia [33,37], dyslipidemia [68], and hypertension [58], have been demonstrated.

The articles that did not find a relationship concluded that the data suggested a possible association, but no statistical significance. They concluded that these results could be explained because of the small sample size or the differences in the ages of the individuals studied [49,69,70].

Regarding the association of oral infections and apical periodontitis with MetS, several mechanisms have been proposed to explain the association between periodontal disease and MetS [55,71] that could also explain the association of the AP with the MetS. In addition, the pleiotropic hormone leptin, produced by adipose tissue, could be a link between oral infections and MetS. Leptin is involved in the regulation of food intake, body weight, energy homeostasis, insulin sensitivity, and lipid metabolism. Leptin has been implicated in the pathogenesis of MetS [72], with high levels of leptin associated with the upregulation of pro-inflammatory cytokines, obesity, insulin resistance, type 2 diabetes, and cardiovascular disease. In both pulp [73] and periapical infections [74] as well as periodontal disease [23], leptin levels are increased. Moreover, leptin receptors are upregulated in inflamed dental pulp [75] and in periapical granuloma [76]. The intestinal levels of leptin increase significantly in rats after 28 days of AP induction [67]. All these results together support the concept that AP and other oral infections could modulate blood leptin levels, which would at least partly explain the association between oral infections and MetS. Finally, the study limitations; if we consider that it is a cross-sectional study, the cause–effect relationship cannot be established and the results must be interpreted with caution.

## 5. Conclusions

Patients who suffer cardiovascular events have poorer oral health and worse periapical condition than the controls. Patients with MetS have worse oral health status, assessed with the modified TDI, but no significant differences have been observed in the periapical status. Two conclusions can be drawn: (1) oral infections, and specifically AP, are significantly associated with cardiovascular events; and (2) total oral inflammatory burden, not AP alone, is associated with MetS.

## Figures and Tables

**Table 1 jcm-09-03205-t001:** Details and definition of the modified total dental index (TDI) used in this study [32].

Type of Disease	Score
**Caries**	
No carious lesions	0
1–3 carious lesions	1
4–7 carious lesions	2
≥8 carious lesions or infected roots or no teeth	3
**Periodontal disease**	
None	0
Gingival pocket 4–5 mm deep	1
Gingival pocket ≥ 6 mm deep	2
Macroscopic pus in gingival pocket	3
**Apical periodontitis**	
None	0
1 tooth or vertical bone pocket, or both	1
2 teeth	2
≥ 3 teeth	3
**Furcation lesions**	
Absent	0
Present	1

**Table 2 jcm-09-03205-t002:** Characteristics and dental status of the patients included both in the study and control groups. Results are expressed as the mean ± standard deviation.

	Total	Study Group	Control Group	*p*
Number of patients	131	83	48	
Age, years	50.0 ± 6.1	50.2 ± 5.9	49.8 ± 6.6	> 0.05
Sex, *n* (%)				
Male	93 (71.0)	58 (69.9)	35 (72.9)	> 0.05
Female	38 (29.0)	25 (30.1)	13 (27.1)	
Diabetes				
Yes	22 (16.8)	15 (18.1)	7 (14.6)	> 0.05
No	109 (83.2)	68 (81.9)	41 (85.4)	
Smoking Habits, *n* (%)				
Yes	107 (81.7)	71 (85.5)	36 (75.0)	>0.05
No	24 (18.3)	12 (14.5)	12 (25.0)	
Body mass index (BMI)	27.4 ± 4.4	28.9 ± 4.5	24.8 ± 2.6	< 0.0001
Metabolic syndrome, *n* (%)				
Yes	71 (54.2)	63 (75.9)	8 (16.7)	< 0.0001
No	60 (45.8)	20 (24.1)	40 (83.3)	
Hypertension, *n* (%)				
Yes	89 (67.9)	70 (84.8)	19 (40)	< 0.001
No	42 (32.1)	13 (15.2)	29 (60)	
Waist circumference, *n* (%)				
Yes	57 (43.5)	47 (56.6)	10 (20)	< 0.001
No	74 (56.5)	36 (43.4)	38 (80)	
Hypertriglyceridemia, *n* (%)				
Yes	71 (54.2)	59 (70.7)	12 (26)	< 0.001
No	60 (45.8)	24 (29.3)	36 (74)	
Low c-HDL, *n* (%)				
Yes	55 (42.0)	47 (56.6)	8 (16)	< 0.001
No	76 (58.0)	36 (43.4)	40 (84)	
High fasting plasma glucose, *n* (%)				
Yes	65 (49.6)	48 (57.6)	17 (36)	0.013
No	66 (50.4)	35 (42.4)	31 (64)	
Number of teeth	23.4 ± 4.5	22.7 ± 4.7	24.6 ± 3.9	> 0.05
No. of root-filled teeth	0.9 ± 1.1	0.9 ± 1.2	0.9 ± 1.0	> 0.05
No. of teeth with apical periodontitis	0.8 ± 1.1	1.0 ± 1.2	0.4 ± 0.7	0.002
At least 1 tooth with apical periodontitis				
Yes	51 (38.9)	39 (47.0)	12 (25.0)	0.013
No	80 (61.1)	44 (53.0)	36 (75.0)	
Total Dental Index (TDI)				
Median	3	4	1	< 0.0000
Interquartile range (IQR, Q1–Q3)	1–5	2–6	0–2	

**Table 3 jcm-09-03205-t003:** Multivariate logistic regression analysis of the influence of the independent variables of age, gender (0 = women, 1 = male), number of teeth, metabolic syndrome (0 = absent, 1 = present), diabetes (0 = absent, 1 = present), smoking habits (0 = non-smoker, 1 = smoker), number of root-filled teeth, and number of teeth with apical periodontitis on the dependent variable “cardiovascular events” (0 = absent, 1 = present).

Variable	B	*p*	OR	Low	High
Age	−0.0349	0.3977	0.9657	0.8906	1.0471
Gender	−0.6956	0.2279	0.4988	0.1610	1.5450
No. of teeth	−0.0628	0.3318	0.9391	0.8273	1.0661
Metabolic syndrome	2.8889	0.0000	17.9736	6.5018	49.6867
Diabetes	−0.0012	0.9985	0.9988	0.2758	3.6172
Smoking habits	0.1852	0.7719	1.2035	0.3440	4.2107
No. of root-filled teeth	−0.3057	0.2058	0.7366	0.4588	1.1827
No. of teeth with AP	0.8489	0.0057	2.3370	1.2806	4.2650

Overall Fit Model: Chi Square = 60.8741; df = 8; *p* = 0.0000; AP: apical periodontitis.

**Table 4 jcm-09-03205-t004:** Multivariate logistic regression analysis of the influence of the independent variables age, gender (0 = women, 1 = male), number of teeth, metabolic syndrome (0 = absent, 1 = present), diabetes (0 = absent, 1 = present), smoking habits (0 = non-smoker, 1 = smoker), number of root-filled teeth, and TDI on the dependent variable “cardiovascular events” (0 = absent, 1 = present).

Variable	B	*p*	OR	Low	High
Age	−0.0418	0.3087	0.9591	0.8849	1.0394
Gender	−0.6272	0.2753	0.5341	0.1731	1.6480
No of teeth	−0.0246	0.7095	0.9757	0.8572	1.1105
Metabolic syndrome	2.5145	0.0000	12.3604	4.5535	33.5519
Diabetes	−0.2870	0.6756	0.7505	0.1957	2.8787
Smoking habits	0.3592	0.5710	1.4322	0.4134	4.9620
No. of root-filled teeth	−0.1532	0.5073	0.8580	0.5455	1.3494
TDI score	0.4273	0.0012	1.5331	1.1827	1.9874

Overall Fit Model: Chi Square = 63.5319; df = 8; *p* = 0.0000; TDI: total dental index.

**Table 5 jcm-09-03205-t005:** Multivariate logistic regression analysis of the influence of the independent variables age, gender (0 = women, 1 = male), number of teeth, diabetes (0 = absent, 1 = present), smoking habits (0 = non-smoker, 1 = smoker), number of root-filled teeth, and number of teeth with apical periodontitis on the dependent variable “metabolic syndrome” (0 = absent, 1 = present).

Variable	B	*p*	OR	Low	High
Age	0.0317	0.3074	1.0322	0.9712	1.0970
Gender	0.1244	0.7640	1.1324	0.5028	2.5503
No. of teeth	−0.0700	0.1196	0.9324	0.8537	1.0183
Diabetes	0.3237	0.5119	1.3822	0.5254	3.6362
Smoking habits	0.5725	0.2395	1.7727	0.6829	4.6018
No. of root-filled teeth	−0.0924	0.6114	0.9117	0.6384	1.3022
No. of teeth with AP	0.1305	0.4908	1.1394	0.7861	1.6516

Overall Fit Model: Chi Square = 7.8241 df = 7; *p* = 0.3484; AP: apical periodontitis.

**Table 6 jcm-09-03205-t006:** Multivariate logistic regression analysis of the influence of the independent variables age, gender (0 = women, 1 = male), number of teeth, diabetes (0 = absent, 1 = present), smoking habits (0 = non-smoker, 1 = smoker), number of root-filled teeth, and TDI score on the dependent variable “metabolic syndrome” (0 = absent, 1 = present).

Variable	B	*p*	OR	Low	High
Age	0.0264	0.4070	1.0268	0.9646	1.0929
Gender	0.0671	0.8759	1.0695	0.4603	2.4847
No of teeth	−0.0242	0.6096	0.9761	0.8894	1.0712
Diabetes	0.0404	0.9377	1.0412	0.3781	2.8670
Smoking habits	0.4630	0.3536	1.5889	0.5974	4.2263
No. root-filled teeth	−0.1433	0.4190	0.8665	0.6121	1.2266
TDI score	0.2878	0.0029	1.3335	1.1035	1.6113

Overall Fit Model: Chi Square = 17.2801; df = 7; *p* = 0.0157; AP: apical periodontitis.

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
