# Peer review of "Relationship between Apical Periodontitis and Metabolic Syndrome and Cardiovascular Events: A Cross-Sectional Study"

_jcm, 2020, doi:10.3390/jcm9103205_

Round 1

Reviewer 1 Report

In this manuscript González Navarro et al investigated the association between apical periodontitis (AP) and CV events. The study is overall interesting, however there are some major flaws that need to be carefully addressed and better explained. Moreover, the reviewer is not familiar with any causal association between AP and systemic inflammation (or CVD) and the Authors did not provide consistent references. It is not clear whether it is a very first observation of this phenomenon, since the Authors refer to other papers that seem to be literature reviews.

1) In the reviewer's understanding, the Authors attempt to assess an association between AP and CV events (or "systemic inflammation"?). However, they did not exlude participants with periodontal diseases (EFP/AAP classification 2018: gingivitis, active periodontitis and stable periodontitis). This is a cornerstone issue that prevents any inference on the selected sample since the association between AP and "systemic inflammation" is biased by the presence of undiagnosed periodontal diseases. 

2) Variables included in the multivariate regression were not reported.

3) Regressions outcome is unclear. Would the Authors analyze CV events longitudinally or cross-sectionally (self-reported)? 

4) The use of the term "oral inflammation" is ambigous since they did not show data regarding this parameter (i.e. cytokines, ROS, redox power, histo...)

Author Response

Thank you very much for your kind review

José López

Reply:

In this manuscript González Navarro et al investigated the association between apical periodontitis (AP) and CV events. The study is overall interesting, however there are some major flaws that need to be carefully addressed and better explained. Moreover, the reviewer is not familiar with any causal association between AP and systemic inflammation (or CVD) and the Authors did not provide consistent references. It is not clear whether it is a very first observation of this phenomenon, since the Authors refer to other papers that seem to be literature reviews.

# Author´s response:

Thanks for your comments. Of course, it has not been shown that there is a causal association between AP and ACVD. Although numerous cross-sectional and some longitudinal studies [15,18,19,20-27] have found an association between oral infections and ACVD, causality has not been demonstrated. To clarify this point, the following sentence has been added: (Page 2)

Despite all these findings, it cannot be considered proven that the association between oral infections and ACVD is causal [10].

1) In the reviewer's understanding, the Authors attempt to assess an association between AP and CV events (or "systemic inflammation"?). However, they did not exlude participants with periodontal diseases (EFP/AAP classification 2018: gingivitis, active periodontitis and stable periodontitis). This is a cornerstone issue that prevents any inference on the selected sample since the association between AP and "systemic inflammation" is biased by the presence of undiagnosed periodontal diseases. 

# Author´s response:

Thanks for your comments and questions. Periodontal status is one of the components of total dental index (TDI) and contribute to oral inflammatory burden. Taking into account that the aim of the study was not only to investigate the prevalence of AP, but also the oral inflammatory burden assessed with TDI, periodontal patients could not be excluded from the study. However, in the multivariate logistic regression analysis, both AP and TDI (including periodontal disease) were included as covariates. Even so, AP remained significantly associated with cardiovascular events. Thus, the results show a significant association between the number of teeth with AP and cardiovascular events (OR = 2.3; p = 0.006), even with the TDI as covariate. The following paragraph has been included in Discussion to comment the possible bias of the results: (Page 9)

However, the results of the present study regarding the association between AP and cardiovascular events could be biased by the presence of periodontal disease. Taking into account that the aim of the study was not only to investigate the prevalence of AP, but also the oral inflammatory burden, assessed with TDI, periodontal patients could not be excluded from the study. Periodontal status is one of the components of total dental index (TDI) and contribute to oral inflammatory burden. Even so, in multivariate logistic regression analysis AP remained significantly associated with cardiovascular events.

2) Variables included in the multivariate regression were not reported.

# Author´s response:

Thanks for your comment. In Results, page 4/12, lines 168-171, and page 4/12, lines 176-179, the independent variables and the dependent variables included in the multivariate logistic regressions analyses are showed. In tables 2 and 3, they are also showed.

3) Regressions outcome is unclear. Would the Authors analyze CV events longitudinally or cross-sectionally (self-reported)? 

# Author´s response:

Thanks for your question. The study is cross-sectional and the multivariate logistic regression analysis has analyzed cross-sectional data. Periapical status, TDI scores and cardiovascular events were evaluated at a single time point. To clarify this point, the expression “occurrence of cardiovascular events” has been changed to “prevalence of cardiovascular events”. 

4) The use of the term "oral inflammation" is ambigous since they did not show data regarding this parameter (i.e. cytokines, ROS, redox power, histo...)

# Author´s response:

Thanks for your comment, it is true what the authors indicate, however we dared to use the term with this ambiguous meaning, since other papers use it in this way, only with a global clinical assessment:

-Brogden KA, Johnson GK, Vincent SD, Abbasi T, Vali S. Oral inflammation, a role for antimicrobial peptide modulation of cytokine and chemokine responses. Expert Rev Anti Infect Ther. 2013 Oct;11(10):1097-113.

-Aarabi G, Schnabel RB, Heydecke G, Seedorf U. Potential Impact of Oral Inflammations on Cardiac Functions and Atrial Fibrillation. Biomolecules. 2018 Aug 1;8(3):66.

-Lim JC, Mitchell CH. Inflammation, pain, and pressure--purinergic signaling in oral tissues. J Dent Res. 2012 Dec;91(12):1103-9.

-Ng KK, Fiani N, Tennant M, Peralta S. Frequency of clinical and radiographic evidence of inflammation associated with retained tooth root fragments and the effects of tooth root fragment length and position on oral inflammation in dogs. J Am Vet Med Assoc. 2020 Mar 15;256(6):687-695.

However, we are aware that other works only refer to this term, when assessing analytical parameters such as for example: Lim JC, Mitchell CH. Inflammation, pain, and pressure--purinergic signaling in oral tissues. J Dent Res. 2012 Dec;91(12):1103-9.

Reviewer 2 Report

Dear authors,

This study investigated the association between apical periodontitis and systemic disease, metabolic syndrome and cardiovascular events. It is interesting topic. However, there are several concerns on the methods and the results.

Statistical analysis seems to be essential to draw the conclusion in this clinical study.

-The number of the control group is almost half of the study group. With my knowledge, it might cause bias when performing statistical analysis. How did you compensate this weakness?

-I am not sure that there is no problem with this multivariate regression. As mentioned in discussion, TDI includes the presence of AP. Therefore, the correlation between TDI and AP is so great that we can't avoid the problem of multicollinearity when both TDI and AP are included as an independent variable. What about figuring out the way to correct the problem?

-The manuscript needs to be revised, especially, in the part of mentioning the aim should be clear. I understand that the aim of this study is to analyze the relation between AP and ACVS, and AP and MetS. However, the primary purpose seems to investigate the association between AP and MetS.

The title should be more specific to represent this research, e.g. Association between apical periodontitis and metabolic syndrome, and apical periodontitis and cardiovascular events: a cross-sectional study

Abstract

- The abstract should represent the overall content of the article. However, because the essential few sentences are omitted in methods and results, it is hard to fully understand this article reading the abstract only. It should provide more explanation, for example, how to obtain the data on MetS.

-What is RFT in line32?

Results

-Table 2. All number in table 2 should be checked. For example, the number of patients having hypertension must be 89, resulting from the 70 from the study group and 19 from the control group.

Discussion

-Line 255-264: I recommend to remove it. It seems to suggest the hypothesis to support the result of this study. I think it is based on the focal infection theory, which is known to the historical concept.

-This study is cross-sectional study, which cannot inform the cause-and-effect relationship. Therefore, it should be cautious to interpret the results.

Author Response

Thank you very much for your kind review.

José López

Reply:

This study investigated the association between apical periodontitis and systemic disease, metabolic syndrome and cardiovascular events. It is interesting topic. However, there are several concerns on the methods and the results.

 Statistical analysis seems to be essential to draw the conclusion in this clinical study.

 -The number of the control group is almost half of the study group. With my knowledge, it might cause bias when performing statistical analysis. How did you compensate this weakness?

# Authors:

Thanks for your comment and question. Taking into account that the size of the control group was almost half of the study group, we used the Welch's t-test (unequal variances t-test), an adaptation of Student's t-test more reliable when the two samples have unequal variances and/or unequal sample sizes. To clarify this point, the sentence in Mat & Met has been modified as follows: (Page 4)

The significance of differences among groups were determined using the χ2 test, the Student´s t-test (Welch's t-test, or unequal variances t-test)

-I am not sure that there is no problem with this multivariate regression. As mentioned in discussion, TDI includes the presence of AP. Therefore, the correlation between TDI and AP is so great that we can't avoid the problem of multicollinearity when both TDI and AP are included as an independent variable. What about figuring out the way to correct the problem?

# Authors:

Thanks for your comment and question. Certainly, as the reviewer explains, the TDI includes periapical lesions, so there may be multicollinearity in multivariate logistic regression analysis. To avoid this problem, the multivariate logistic regression analysis has been performed separately for each of the two variables: number of teeth with AP and TDI. This has been done both in the case of cardiovascular events as a dependent variable, and in the case of metabolic syndrome. Therefore, two new tables have been included.

 -The manuscript needs to be revised, especially, in the part of mentioning the aim should be clear. I understand that the aim of this study is to analyze the relation between AP and ACVS, and AP and MetS. However, the primary purpose seems to investigate the association between AP and MetS.

# Authors:

We are sorry we apologize, but we do not understand very well what does not seem right to you

-The title should be more specific to represent this research, e.g. Association between apical periodontitis and metabolic syndrome, and apical periodontitis and cardiovascular events: a cross-sectional study

# Authors:

Thanks for your comment.  We have changed the title of the pape: (Page 1)

Relationship between apical periodontitis and metabolic syndrome and cardiovascular events: a cross-sectional study

Abstract

-The abstract should represent the overall content of the article. However, because the essential few sentences are omitted in methods and results, it is hard to fully understand this article reading the abstract only. It should provide more explanation, for example, how to obtain the data on MetS

# Authors:

Thank you very much for the comment. We have changed the part in the abstract: (Page 1)

Diagnosis of Mets was made by meeting 3 or more American Heart Association Scientific Statement components.

-What is RFT in line32?

# Authors:

We have detailed the initials of RFT: (Page 1)

...root-filled teeth (RFT)

Results

-Table 2. All number in table 2 should be checked. For example, the number of patients having hypertension must be 89, resulting from the 70 from the study group and 19 from the control group.

# Author´s response:

Thanks for your comments and suggestion. All the figures in Table 2 have been revised and those that were erroneous have been corrected. (Page 5)

Discussion

-Line 255-264: I recommend to remove it. It seems to suggest the hypothesis to support the result of this study. I think it is based on the focal infection theory, which is known to the historical concept.

# Author's response:

Following your suggestion, we have removed the entire paragraph and readjusted the references. Similarly, we have added a paragraph in response to the suggestions indicated by the reviewer number three and have readjusted the references.
Removed:

(Page 9)

Although it is not the objective of this work, in order to guarantee the biological plausibility and scientific coherence of the results of this study, it is convenient to consider the possible biological mechanisms explaining the observed association between both oral inflammatory burden and endodontic infection with cardiovascular events and ACVD. These mechanism could be the following [8,10,62]: 1) transient bacteraemia, with spreading of infection from the oral cavity to the vascular endothelium, favouring the development of atherosclerotic plaque [63,64]; 2) stimulation of systemic inflammation by the periapical inflammatory response, increasing  the inflammatory burden related to the onset of atherosclerosis [65]; 3) causing early endothelial dysfunction with reduced endothelial flow reserve [28]; 4) higher production of reactive oxygen species, strongly involved in the pathology of atherosclerosis [66].

These references were removed (Previous numbering)

  1. Hajishengallis, G.; Lambris, J.D. More than complementing Tolls: complement-Toll-like receptor synergy and crosstalk in innate immunity and inflammation. Immunol. Rev. 2016, 274, 233-244.
  2. Dorn, B.R.; Harris, L.J.; Wujick, C.T.; Vertucci, F.J.; Progulske-Fox, A. Invasion of vascular cells in vitro by Porphyromonas endodontalis. Int. Endod. J. 2002, 35, 366–371.
  3. Hernández-Ríos, P.; Pussinen, P.J.; Vernal, R.; Hernández, M. Oxidative stress in the local and systemic events of apical periodontitis. Front. Physiol. 2017, 8, 869.
  4. Cotti, E.; Mercuro, G. Apical periodontitis and cardiovascular diseases: previous findings and ongoing research. Int. End. J. 2015, 48, 926–932.
  5. Minczykowski, A.; Woszczyk, M.; Szczepanik, A.; Lewandowski, L.; Wysocki, H. Hydrogen peroxide and superoxide anion production by polymorphonuclear neutrophils in patients with chronic periapical granuloma, before and after surgical treatment. Clin Oral Investig. 2001, 5, 6–10.

Added: (Page 9)

The articles that not found a relationship concluded that the data suggested a possible association, but not statistical significative. They concluded that these results could be explained because of the small sample size or the differences in the ages of the individuals studied (42,62,63).

The following references have been added: (Page 11 & 12)

  1. Mattila, K.J.; Asikainen, S.; Wolf, J.; Jousimies-Somer, H.; Valtonen, V.; Nieminen. M. Age, dental infections, and coronary heart disease. J. Dent, Res. 2000, 79, 756-760.
  2. Friedlander, A.H.; Sung, E.C.; Chung, E.M.; Garrett, N.R. Radiographic quantification of chronic dental infection and its relationship to the atherosclerotic process in the carotid arteries. Oral Surg. Oral Med. Oral Pathol. Oral Radiol. Endod. 2010, 109, 615-621.

-This study is cross-sectional study, which cannot inform the cause-and-effect relationship. Therefore, it should be cautious to interpret the results.

#Author's response:

Following your suggestions, we have added a study limitation: (Page 9)

Finally, the study limitations. If we consider that it is a cross-sectional study, the cause-effect relationship cannot be established and the results must be interpreted with caution.

Reviewer 3 Report

The present paper is well written and organized.

I have few minor suggestions:

  1. About metabolic syndrome and cardiovascular diseases, please refer to: DOI: 1186/1743-7075-9-88 (Marchetti et al.)
  2. Please, include in the inclusion criteria patients that agreed to do blood analyses for MetS diagnosis. Did control subjects agreed?
  3. Please, report reference for the Spanish population used to assess sample size
  4. Among the results, calculate the Odds ratio for subjects with SG to be affected by Mets

Reviewer 4 Report

This well performed and clearly reported study adds further observations to corroborate the previously connections and associations between oral inflammatory lesions  (OIL) and cardiovascular disease (CVD). The study focus in particular on the contribution to this association that apical periodontitis (AP) may have. The study merits publication. However there are some methodological issues and difficulties that should be commented/discussed by authors.

  • For the diagnosis of AP the observers were calibrated and used the PAI score. This method is widely accepted within endodontology but is originally adapted and validated on intra oral radiographs. An explanation or comment is needed.
  • One of the advantages of the PAI-score is that is a provide a possibility to provide a relative judgement about the histological features of the apical status on 5 steps cardinal scale where 1 denotes a healthy situation without any inflammation while 5 denotes the most severe deviation from a healthy situation. However, the authors chose to dichotomize their findings (1,2 or 3,4,5). Why? Comment, explanation, please.
  • In the M&M the authors in detail describe how the radiographic assessment was performed and the results of observers’ calibration. Good. Because it is a difficult task to diagnose AP only from OPG’s. However, there are other dental diagnoses (carious lesions, periodontal disease and furcation lesions) these categories of OILs are also classified to different stages of disease (for example no carious, 1-3 carious lesions, 4-7 lesions). What about the reliability and validity of these data?
  • The authors should, I think, also discuss the possible reasons why different studies on this or other manifestations of CVD sometimes suggest an association between AP and CVD and sometimes not.
  • Finally, I think there are reasons to from both from a logical point of view as well as from the complexity of the issues here investigated and the limitations of any observational study to doubt the validity of the implicit in the discussion and conclusion that since AP is associated with CVD as is MetS while AP is not associated with MetS, “oral infections, and specifically AP, are significantly associated with cardiovascular disease”. Could it be because of a TypeII error? A unknown confounder? Reconsider the conclusion?

Author Response

Thank you very much for your kind review.

José López

Reply:

This well performed and clearly reported study adds further observations to corroborate the previously connections and associations between oral inflammatory lesions  (OIL) and cardiovascular disease (CVD). The study focus in particular on the contribution to this association that apical periodontitis (AP) may have. The study merits publication. However there are some methodological issues and difficulties that should be commented/discussed by authors.

# Author´s response:

Thank you very much for your kind and favorable comments.

For the diagnosis of AP the observers were calibrated and used the PAI score. This method is widely accepted within endodontology but is originally adapted and validated on intra oral radiographs. An explanation or comment is needed.

#Authors:

Thanks for your comment and suggestion. The answer needs to discuss two main points: 1) The use of panoramic radiographs to assess periapical status and 2) The use of PAI score system with panoramic radiographs. 1) In relation to the use of panoramic radiographs to assess periapical status, both periapical radiography and panoramic radiography have been used to assess the periapical status both in experimental (Flint et al. 1998) and epidemiological studies (De Moor et al. 2000, Lupi-Pegurier et al. 2002, Dugas et al. 2003, Ridao-Sacie et al. 2007, Segura-Egea et al. 2010, López-López et al. 2011, López-López et al. 2013, Castellanos-Cosano et al. 2013, Sánchez-Domínguez et al. 2015, Poyato-Borrego et al. 2020). Although an underestimation of periapical lesions has been reported when panoramic radiography was used to assess the periapical status (Eriksen & Bjertness 1991), the difference with periapical radiography was not statistically significant (Muhammed et al. 1982, Ahlqwist et al. 1986). On the contrary, other studies have shown that panoramic radiographs achieve significantly higher percentages of teeth with periapical pathosis (Nishikawa et al. 2000, Ríos-Santos et al. 2010). Furthermore, the fact that all teeth can be seen on one panoramic radiograph, the relatively low exposure to ionizing radiation (considering that the radiation dose from a panoramic radiograph approximately corresponds to 2-4 intraoral radiographs, the dose reduction is 40-50%, as referred Molander et al. 1995), the convenience of panoramic radiographs and the speed with which they can be obtained are advantageous when compared with full-mouth periapical radiographs (Gulsahi et al. 2008). Consequently, panoramic radiographs are a highly viable tool to implement studies in a rapid fashion (Gutmann et al. 2009). So,  a lot of epidemiological studies have been carried out using panoramic radiographs (De Cleen et al. 1993, Marques et al. 1998, De Moor et al. 2000, Lupi-Pegurier et al. 2002, Dugas et al. 2003, Skudutyte-Rysstad & Eriksen 2006, Ridao-Sacie et al. 2007, Willershausen et al. 2009, Segura-Egea et al. 2010, López-López et al. 2011, Peršić et al. 2011, Matijević et al. 2011, Kamberi et al. 2011, Gomes et al. 2012, López-López et al. 2012, Marotta et al. 2012, Tolias et al. 2012, López-López et al. 2013, Castellanos-Cosano et al. 2013, Jersa & Kundzina 2013, Sánchez-Domínguez et al. 2015, Poyato-Borrego et al. 2020). In conclusion, panoramic radiography is a useful and valid method to assess periapical status, particularly in epidemiological studies.

2) Respect to the use of PAI score system with panoramic radiographs, it must be taken in mind that in the evaluation of the apical periodontium, bone density changes present in radiographs are the most consistent feature of the presence, progression or resolution of periapical inflammation. Although there seemed to be no standard criteria for the registration of apical periodontitis in epidemiological surveys, either for periapical radiographs or panoramic radiographs, the ‘periapical index’ (PAI) scoring system (Orstavik et al. 1986), based on reference radiographs with verified histological diagnoses published originally by Brynolf (1967), has been applied to epidemiological (Kirkevang et al. 2000; Jiménez-Pinzón et al. 2004; Segura-Egea et al. 2008; López-López et al. 2011; Castellanos-Cosano et al. 2013, Sánchez-Domínguez et al. 2015, Poyato-Borrego et al. 2020) and clinical comparative studies of treatment outcome (Kirkevang et al. 2001, Segura-Egea et al. 2005). The possibility of comparisons among studies carried out with calibrated observers makes this system attractive (Huumonen & Orstavik 2002). PAI provides an ordinal scale of five scores ranging from ‘healthy’ to ‘severe periodontitis with exacerbating features”. Although  PAI was first described for periapical radiographs (Ørstavik et al. 1986), a lot of epidemiological studies have used PAI for panoramic radiographs alone (Skudutyte-Rysstad & Eriksen 2006, López-López et al. 2011, Peršić et al. 2011, Matijević et al. 2011, Kamberi et al. 2011, Tolias et al. 2012, López-López et al. 2013, Castellanos-Cosano et al. 2013, Jersa & Kundzina 2013, Sánchez-Domínguez et al. 2015, Poyato-Borrego et al. 2020) or in combination with periapical radiographs (Eriksen & Bjertness 1991, Weiger et al. 1997, Eriksen et al. 1998, Sidaravicius et al. 1999, Dugas et al. 2003). In conclusion, PAI can be used to assess periapical status in panoramic radiographs in order to better standardize the evaluations and to allow the comparison with the result of other investigators.

To clarify these points in the manuscript, the following paragraph has been added in the Discussion section: (Page 8) “Although there may be an underestimation of lesions when panoramic radiography was used [38], the difference with periapical radiography is not statistically significant [39]. Both periapical and panoramic radiography can be used to assess periapical status, but the fact that panoramic radiography shows all teeth, together with the low exposure to ionizing radiation [39], the convenience of panoramic radiographs, and the speed with which they can be obtained are advantageous compared to periapical full-mouth radiographs [40]. On the other hand, although PAI has been described for periapical radiographs [7], it has been used by numerous epidemiological studies in which the periapical status is assessed with panoramic radiographs [41,42]. The possibility of comparisons between studies carried out with calibrated observers makes this system attractive [43] to better standardize the evaluations and allow comparison with the results of other researchers.

The following references have been added: (Pqge 11 & 12)

  1. Eriksen HM, Bjertness E. Prevalence of apical periodontitis and results of endodontic treatment in middle-aged adults in Norway. Endod Dent Traumatol 1991;7:1-4.
  2. Molander B, Ahlqwist M, Gröndal H-G. Panoramic and restrictive intraoral radiography in comprehensive oral radiographic diagnosis. Eur J Oral Sci 1995;103:191-8.
  3. Gulsahi K, Gulsahi A, Ungor M, Genc Y. Frequency of root-filled teeth and prevalenceof apical periodontitis in an adult Turkish population. Int Endod J 2008;41:78–85
  4. Gomes MS, Hugo FN, Hilgert JB, Padilha DM, Simonsick EM, Ferrucci L, Reynolds MA. Validity of self-reported history of endodontic treatment in the Baltimore Longitudinal Study of Aging. J Endod. 2012; 38:589-93.
  5. Skudutyte-Rysstad R, Eriksen HM. Endodontic status amongst 35-year-old Oslo citizens and changes over a 30-year period. Int Endod J 2006;39:637-42.
  6. Huumonen S, Ørstavik S. Radiological aspects of apical periodontitis. Endodontic Topics 2002;1:3 – 25.

On the other hand, although PAI score system was first described for periapical radiographs [26], it has been used in many epidemiological studies for panoramic radiographs [22,24,27,36,37,44,45]. The possibility of comparisons among studies carried out with calibrated observers makes this system attractive [46] in order to better standardize the evaluations and to allow the comparison with the result of other investigators.

One of the advantages of the PAI-score is that is a provide a possibility to provide a relative judgement about the histological features of the apical status on 5 steps cardinal scale where 1 denotes a healthy situation without any inflammation while 5 denotes the most severe deviation from a healthy situation. However, the authors chose to dichotomize their findings (1,2 or 3,4,5). Why? Comment, explanation, please.

# Author´s response:

Thanks for your comment and question. Although the PAI scoring system establishes 5 possible periapical status scores, to perform the multivariate logistic regression analysis it is necessary to dichotomize the variable. On the other hand, being an index it must be treated as a qualitative variable. Therefore, most epidemiological studies also perform this dichotomization.

In the M&M the authors in detail describe how the radiographic assessment was performed and the results of observers’ calibration. Good. Because it is a difficult task to diagnose AP only from OPG’s. However, there are other dental diagnoses (carious lesions, periodontal disease and furcation lesions) these categories of OILs are also classified to different stages of disease (for example no carious, 1-3 carious lesions, 4-7 lesions). What about the reliability and validity of these data?

# Author´s response:

Thank you very much for your comments. The authors of the paper are clinicians, some with more than 25 years of experience and the Mattila index is numerical and that makes its reproducibility relatively easy. We have to admit that we have not carried out a specific calibration on this index.

The authors should, I think, also discuss the possible reasons why different studies on this or other manifestations of CVD sometimes suggest an association between AP and CVD and sometimes not.

# Author´s response:

Thank you very much for the comment. We have changed the part in the discussion (Page 9)

The articles that not found a relationship concluded that the data suggested a possible association, but not statistical significative. They concluded that these results could be explained because of the small sample size or the differences in the ages of the individuals studied [48,68,69]

Finally, I think there are reasons to from both from a logical point of view as well as from the complexity of the issues here investigated and the limitations of any observational study to doubt the validity of the implicit in the discussion and conclusion that since AP is associated with CVD as is MetS while AP is not associated with MetS, “oral infections, and specifically AP, are significantly associated with cardiovascular disease”. Could it be because of a TypeII error? A unknown confounder? Reconsider the conclusion?

# Author´s response:

Thanks for your comments and questions. To discuss this points, the following paragraphs has been added in Discussion:

Page 8)

However, the results of the present study regarding the association between AP and cardiovascular events could be biased by the presence of periodontal disease. Taking into account that the aim of the study was not only to investigate the prevalence of AP, but also the oral inflammatory burden, assessed with TDI, periodontal patients could not be excluded from the study. Periodontal status is one of the components of total dental index (TDI) and contribute to oral inflammatory burden. Even so, in multivariate logistic regression analysis AP remained significantly associated with cardiovascular events.

(Page 9)

Given that endodontic infection is included in the TDI score, these results are not conclusive regarding the association or not of AP with MetS, but suggest that both AP and MetS are independently associated with cardiovascular events.

Round 2

Author Response

Reviewer 1

(x) Extensive editing of English language and style required

Response:

Find all fixes with change control in the new uploaded file

Reviewer 4 Report

I think the authors have given satisfactory answers to my questions and amde amendments as far it is possible. No further comments.

Author Response

(x) Extensive editing of English language and style required

Response:

Find all fixes with change control in the new uploaded file

Many thanks agaiin
